# Assessing the Differential Abundance of Maternal Circulating MicroRNAs or Interferon-Stimulated Genes with Early Pregnancy

**DOI:** 10.3390/genes14081532

**Published:** 2023-07-27

**Authors:** Andrea N. DeCarlo, Joseph Parrish, Jasmine D. Quarles, Nathan M. Long, Scott L. Pratt

**Affiliations:** Department Animal and Veterinary Sciences, Clemson University, Clemson, SC 29634, USA

**Keywords:** microRNA, interferon-stimulated genes, RT-qPCR, pregnancy

## Abstract

Interferon-stimulated genes (ISG) and microRNA (miRNA) present in maternal circulation have been reported to be diagnostic of pregnancy in cattle prior to day (d)30 of gestation. The objective of this study was to assess specific ISG and miRNA abundance on d 18 of gestation. Cattle were subjected to estrous synchronization and artificially inseminated to a single Angus sire. At time of insemination (d 0) and d 18 post-insemination, blood was collected and total RNA isolated. Differential abundance (DA) in specific ISG and miRNA between d 0 and d 18 samples in pregnant (*n* = 10) and open (*n* = 10) cows were assessed via RT-qPCR. The relative Ct values were normalized using abundance of cyclophilin or the geometric mean of specific miRNA for the ISG and miRNA genes of interest, respectively. The DA of the ISG were increased due to pregnancy (*p* < 0.05); however, there was no expected day of gestation by pregnancy interaction. Relative abundance of Bta-miR-16 increased on d18 regardless of pregnancy status (*p* < 0.05). None of the miRNA evaluated in this study were associated with pregnancy status. These data indicate that certain ISG may serve as early indicators of pregnancy in cattle, but abundance of the miRNA does not.

## 1. Introduction

High rates of pregnancy loss have been observed when using assisted reproductive techniques such as IVF [1,2,3] and somatic cell nuclear transfer coupled to embryo transfer [4,5]. Much effort and money would be saved if embryos could be effectively evaluated for developmental competence prior to transfer. While efforts have been made to identify markers for developmentally competent embryos and normal developing pregnancies, no effective screening procedures based on biomarkers have been developed. Interestingly, recent research indicated that markers present in maternal circulation may be diagnostic of pregnancy prior to day 28 of gestation in cattle [6,7,8,9,10,11], and may be diagnostic of developmental competence of the embryonic period of fetal development [8].

It has also been argued that determination of pregnancy prior to the day 28 to day 30 of gestation would be beneficial to producers enabling quick financial decisions. Pregnancy recognition in cattle occurs around day 16 of gestion [12]. This process, known as maternal recognition of pregnancy, is controlled in ruminants by the conceptus, which releases interferon tau (IFNtau) from the trophoblast cells [13,14], binding to receptors present in the uterine endometrium and altering PGF2a release [13,15], thus blunting corpus luteum demise. Interestingly, IFNtau of conceptus origin has been reported to impact gene expression in immune cells circulating in the maternal system. If maternal immune cells react to IFNtau, the detection of interferon-stimulated genes (ISG) could be useful in detecting pregnancies earlier than day 28–30 of gestion [16,17,18]. Several studies showed that ISG abundance determined via RT-qPCR, in particular abundance of ISG15, MX1 and OAS1 from cells isolated from the buffy-coat were higher in pregnant animals compared to non-pregnant cows and could be used as an indicator of early pregnancy detection [10,16,17,18].

Other biomolecules are also being evaluated as determinants of pregnancy and as markers for competence of early embryonic development. MicroRNA (miRNA) are small non-coding RNA produced within the cell and can inhibit transcription of mRNA by association with the 3′ UTR, suppressing translation and eventually targeting the mRNA for degradation within the cell [19,20]. miRNA are also known to be released from cells into body fluids via microvesicle release, exosome release, apoptotic body release or as protein-bound miRNA [21,22], and this observation led many researchers to examine the usefulness of circulating miRNA as markers for specific diseases for specific physiological status [21,23,24,25,26,27]. Several studies indicated that miRNA may be an effective marker for pregnancy and for delineating normal embryonic development versus abnormal embryonic development resulting in pregnancies at risk of termination [6,7,8,9,28,29,30,31]. However, it is becoming apparent based on previous reports that standardization of procedures for the assessment of specific miRNA associated with specific physiological states is required [32,33,34]. Regardless if miRNA are isolated from serum, plasma or exosomes, care should be taken to assess the degree of cellular lysis in samples as this could alter any interpretation of relative abundance of specific miRNA. The objective of this study was to verify (1) relative ISG mRNA abundance in maternal serum at d 18 of gestation and evaluate their association with pregnancy at d 30 of gestation and (2) determine if specific miRNA previously reported to be present in maternal circulation and associated with pregnancy status differed in relative abundance between d 0 and d 18 of gestation using two different RNA isolation procedures. To achieve these objectives, we utilized two procedures for the isolation of RNA for determining relative abundance for the expression of either ISG or miRNA in maternal circulation at d 0 and d 18 of gestation to evaluate the impact of sample isolation procedures on the target relative abundance. Due to the remote location of the cattle available for use in this study, it was decided to utilize the PAXgene blood RNA kit (Pre AnalytiX, Hombrechtikon, Switzerland) as it would allow for stable transport from the animal facility to labs capable of processing the samples for total RNA isolation. These samples would contain total RNA from all blood cells and miRNA in maternal circulation. We also decided to evaluate total circulating miRNA in serum samples and isolated RNA using the Qiagen miRNeasy (Qiagen, Ann Arbor, MI, USA).

## 2. Materials and Methods

### 2.1. Animals and Experimental Design

All animal procedures were approved by Clemson University Animal Care and Use Committee (AUP #2018-048 and #2019-023), and all animals were housed at a single location, and subjected to all treatments within the time in the same year. Primi- and multiparous Angus and Angus crossbred cows (n = 80) greater than 45 days postpartum were synchronized using the C0-synch plus CIDR protocol (Figure 1). All animals were bred AI 60 to 66 h post-CIDR removal. Blood samples were obtained on each animal via venipuncture at TAI (day = 0), d 18 post-AI, and d 30 post-AI and blood processed using two different procedures. Blood processed for serum was allowed to clot, stored at 4 °C overnight, then processed for serum by centrifugation at 2000× *g* for 10 min. The serum was collected and stored at −80 °C until used in analysis. Prior to analysis, serum was thawed, subjected to centrifugation and subjected to spectrophotometry to determine the degree of hemolysis [33,34], as hemolysis release RBC miRNA, which could yield misleading results. Only samples within the range reported by Calcatera et al., 2018, were used [35]. Blood samples (2.5 mL) that were to be utilized for the isolation of total RNA, circulating RNA and RNA present in cells an extracellular vesicles, were placed into the PAX blood collection tubes (PreAnalytiX, Hombrechtikon, Switzerland). The samples were processed according to the manufacturer’s guidelines and stored at −80 °C until used for RNA isolation.

### 2.2. Pregnancy Detection

All animals were subjected to pregnancy detection. Pregnancy was determined by an animal being diagnosed as pregnant or open using trans-rectal ultrasonography and the BioPryn ELISA (BioTracking, Inc., Moscow, ID, USA), which detects pregnancy specific protein b (PSPb) [36]. Briefly, transrectal ultrasonography was conducted using an Aloka 500 unit fitted with a 7.5 MHz probe. Pregnancy was verified visually with the presence of a conceptus and amniotic fluid. BioPRYN ELISA was conducted per manufacturer’s directions. A subset of 10 pregnant and 10 open animals at day 30 of gestation were utilized for experiments. The subset of animals were all of similar age and body condition, isolated serum samples were free of hemolysis, and pregnancy verified by both ultrasonography and the BioPryn ELISA.

### 2.3. RNA Isolation

RNA was isolated either from whole blood collected into PAX blood collection tubes or from serum. Total RNA was isolated from blood samples using the PAXgene Blood RNA Kit (PreAnalytiX, Hombrechtikon, Switzerland) as described by the manufacturer. Quantity and quality of total RNA isolated were assessed using spectrophotometry (Nanodrop, ThermoFisher, Waltham, MA, USA) and the Agilent 6000 Nano Chip kit on the Agilent 2100 (Agilent Technologies, Santa Clara, CA, USA). Only samples with a RIN of greater than or equal to 7 were used in further analysis. Total RNA was isolated from serum using the Qiagen miRNeasy kit (Qiagen, Ann Arbor, MI, USA) per manufacturer directions. Concentration and purity of RNA was determined using the Nanodrop 1000 (Thermo Fisher, Waltham, MA, USA) and quality assessed using the Agilent 2100 small RNA kit.

### 2.4. RT-qPCR for INF Stimulated Genes

RNA isolation from samples using the PAXgene blood RNA kit would contain blood cells, including white blood cells, in the lysate producing total RNA. These samples were utilized in RT-qPCR to determine the relative abundance of 3 INF stimulated genes [17]. Genes, accession number, primer sequence, product size, R2, slope and primer efficiency are given in Table 1. Primers were as described by Green et al., 2010. Primers were validated first by performing end-point RT-PCR of PAXgene isolated total RNA followed by slab gel electrophoresis of the products to verify product size and to determine the presence or absence of product as previously reported. Additionally, all products were subjected to dideoxy sequencing to verify product identity (Genewiz, South Plainfield, NJ). Primer efficiencies were determined by performing PCR amplifying increasing amounts of cDNA generated from PAXgene-isolated total RNA for 35 cycles and graphing the log (sample concentration) against the respective Ct value for each point. Linear regression analysis generated the slope and primer efficiency was calculated using the formula Efficiency (%) + (100^1/slope^) × 100. For RT-qPCR, an equal mass of each RNA sample (100 ng) was subjected to reverse transcription using qScript cDNA synthesis kit (Quanta Bio, Veverly, MA, uSA). qPCR was conducted using 5 ng of resulting cDNA with PerfeCTa SYBR Green FastMix (Quanta Bio, Beverly, MA, USA) in duplicate for each sample for the mRNA for MX2, Oas1 and Isg11. Data were normalized using cyclophilin [17]. Relative abundance was determined using the following formula: gene expression ratio = (E _gene of interest_) ^ΔCt gene of interest^/(E cyclophilin) ^ΔCt cyclophilin^ where E is the efficiency of each primer set [37]. Receiver operator characteristic curves were generated for each ISG [38].

### 2.5. RT-qPCR for Specific miRNA

To determine the abundance of specific miRNA in maternal serum at d0 or d18 of gestation, Taqman (Thermo Fisher, Waltham, MA) PCR was conducted on samples extracted using the PAXgene method (contained cells) or the Qiagen miRNeasy method (no cells present). The assay ID is given in the table and checked against Bos taurus miRNA sequences. The miRNA evaluated were reported to be present in maternal serum [6,8,29,39]. The miRNA chosen for analysis had either been reported to be present or differentially abundant in bovine maternal circulation ([6,8,30]: Table 2) or have been reported to be present in bovine serum and excellent to normalize expression [39]. All miRNA were evaluated using BestKeeper analysis, and bta-miR-127, -128 and -9 3 were identified as acceptable for normalization of the data. Briefly, 10 ng of total RNA isolated from whole blood or serum was subject to RT-qPCR for Taqman primers (Table 2) and conducted as described by the manufacturer. To determine differential abundance of gene expression, the Vandesomplele method was used as shown in the formula: relative gene expression = the RQ _gene of interest_/Geometric mean (RQ _of reference genes_). RQ is defined as the primer efficiency ^ΔCt^ [40]. Receiver operator characteristic curves were generated for each target miRNA [38].

### 2.6. Statistical Analysis

ISG RT-qPCR data were subjected to analysis using ANOVA (JMP, SAS Institute, Cary, NC, USA). Model fixed effects were pregnancy status determined at day 30 of gestation (Open or Pregnant), day of gestation (d0 or d18) or their interaction. RNA isolation and miRNA relative differences in abundance data were evaluated where model fixed effects were pregnancy status determined at day 30 of gestation (Open or Pregnant), day of gestation (d0 or d18), RNA isolation method or their interactions. Student’s t-test among LS means for RNA quality, INF stimulated gene abundance and miRNA abundance was used to determine significance at the *p* < 0.05.

## 3. Results

### 3.1. RNA Isolation

RNA was isolated from whole blood using PAXgene blood RNA kit that isolated RNA from all components of blood including red and white blood cells or serum (no cellular RNA) using the Qiagen miRNeasy procedure. The RNA profiles for RNA isolated using the PAXgene or RNeasy procedures are shown in Figure 2. The use of the PAXgene procedure has not been well documented in cattle and, so, all RNA was evaluated for quantity and quality of RNA obtained from both pregnant and open cows (Figure 2). Within the RNA isolation method, no difference was observed for quantity or quality of RNA due to the day of gestation (Figure 2). No difference was observed for RIN values for RNA isolated using the PAXgene procedure. 

### 3.2. RT-PCR Analysis of INF Stimulated Genes

End-point PCR analysis was conducted on all 20 samples with a representative gel shown in Figure 3 (Cyclophilin not shown). All products were of the expected size and identity verified by dideoxy sequencing and present in both open and pregnant animals on d 0 and d 18. RT-qPCR of the ISG showed increased abundance due to pregnancy status (*p* < 0.05); however, only ISG15 exhibited a trend for a pregnancy by day interaction (*p* < 0.1).

### 3.3. RT-qPCR Analysis of miRNA Abundance

The relative abundance for Bta-let-7a, miR-15b, miR-25 and miR-26 were highest in the PAXgene RNA samples (*p* < 0.05); however, only Bta-miR-25 tended to have increased abundance for isolation methods by day interaction (*p* < 0.1) (Table 3). Method of RNA isolation did not affect relative abundance of miR-16a. Relative abundance of miR-16a was higher in samples from d 0 compared to d 18 (*p* < 0.05) and tended to be elevated in the d 0 samples of cows that were open at d 30 of gestation (*p* < 0.1). No miRNA evaluated exhibited increased abundance or tended to exhibit an elevated abundance in d 0 or d 18 samples from cows verified as pregnant at d 30 of gestation. 

### 3.4. Receiver Operator Characteristic Curves

Receiver operator characteristic curves (ROC) were generated for each target ISG and miRNA evaluated to determine their usefulness as a diagnostic tool for accurately predicting pregnancy. ROC curves were generated for the ddCT of each ISG are shown in Figure 4. The area under the curve (AUC) for the ddCT values were 0.41, 0.73 and 0.79 for Isg15, MX2 and Oas1, respectively. In contrast, all AUC values for the relative gene abundance of miRNA values evaluated were under 0.50 (curves not shown) and, therefore, not predictive of pregnancy status.

## 4. Discussion

### 4.1. miRNA as Markers for Pregnancy Detection and at Risk Pregnancies

The reproductive efficiency when using assisted reproductive technologies, such as AI or ET, could be increased if reliable methods to detect developmentally incompetent embryos or at risk pregnancies versus developmentally normal pregnancies early in gestation were available. These detection methods would allow producers to make management decisions earlier in the process saving time and money. Circulating miRNA are present in biological fluids [21,26] and have been proposed to be useful as biomarkers for disease and physiology [20,23,25,27,41]. Circulating miRNA have also been suggested to have biological functions as a way for cell communication and gene regulation [22,42]. Interestingly, miRNA in circulation can be associated with extra-cellular vesicles or ribonucleoprotein complexes with heterogeneity found in vesicle size, proteins associated with vesicles or protein components of the ribonucleoprotein complexes [43,44]. In ruminants, there is currently no method to identify the source of production of circulating miRNA present in maternal serum. Several studies indicated that circulating maternal miRNA [6,7,8,9,29,30,31] or the expression of ISG in circulating maternal cells [10,16,17,18,45,46,47] may be diagnostic of pregnancy or pregnancies that are at risk. Pregnancy recognition in cattle occurs around day 16 of gestion [12] and is driven by the conceptus production of IFNtau from the trophoblast cells. The thought is that IFNtau of conceptus origin stimulates ISG abundance in immune cells present in the maternal circulation [10,17]. Typically, maternal blood samples are obtained, plasma isolated by centrifugation and the buffy-coat used to isolate total RNA from circulating blood cells for analysis [17]. If maternal cells react to IFNtau, the detection of ISG could be useful in detecting pregnancies earlier than day 28–30 of gestation. 

Our long-term goal is to utilize maternal miRNA as markers for pregnancy health in ruminants when using either traditional or assisted reproductive technologies such as ET or SCNT. Others investigators have similar interests. Some focused on early embryonic loss and identified miRNA associated with pregnancy or pregnancy loss [6,8,29,30,31]. Unfortunately, there is inconsistency with reports of miRNA identified and which miRNA may be differentially abundant due to pregnancy. Circulating miRNA has been evaluated from samples that were isolated from serum [30], plasma [6,29], or isolated exosomes [8,31]. Using exosomes as the source of total RNA for evaluation is an excellent choice with the exceptions of (1) source of the exosomes cannot be determined, (2) currently, miRNA association with any specific size of extra-cellular vesicle cannot be determined, and (3) it discounts the large population of miRNA associated with ribonucleoprotein complexes in circulation [43]. Therefore, we utilized RNA isolated from whole blood or serum to assess miRNA abundance. It was hoped that the abundance and presence of specific ISG would assist us to identify animals that were pregnant or open at d 18 of gestation based on previous reports. The PAXgene blood RNA kit (PreAnalytiX, Hombrechtikon, Switzerland) was used to isolate RNA from whole blood as it would allow for stable transport from the animal facility to labs capable of processing the samples for total RNA isolation. Efforts to carefully characterize all RNA to ensure quality of the samples are shown in Figure 2. 

### 4.2. Assessment of Isolated RNA Quality

The PAXgene blood RNA kit gave high-quality samples with distinct 28 s and 18 s bands, 260/280 ratios of over 2.0 and RIN values > 7.5. All these parameters indicate quality total RNA for the subsequent use in RT-qPCR for ISG and specific miRNA. Previous reports determining the relative abundance of ISG were performed by the isolation of white blood cells [17]. The method used here isolated total RNA from whole blood that would include all cells in circulation as well as cell free RNA. In addition, analysis of total RNA isolated from serum clearly identifies the presence of small RNA in the expected size range for miRNA and the results are consistent with our previous studies characterizing serum RNA from bulls and boars [35,48]. There was a significant difference in the concentration of total RNA isolated in the d18 samples compared to d0 when using the PAX gene blood RNA isolation kit; however, no difference in total yield was observed. The total amount of RNA obtained per mL of blood using the PAXgene Blood RNA kit is within parameters reported by others using RNAzol (6.7 to 22 mg/mL whole blood) [49]. No RIN values were reported; however, the 28S:16S ratio observed in those samples were low compared to other tissues using the same procedures. As expected, RNA isolated from serum exhibited lower mass, concentration and A260/280 ratio. Basically, total blood RNA was of acceptable quantity and quality for use in RT-qPCR. In addition, the total RNA isolated from blood were within ranges previously reported for cattle.

### 4.3. Relative Abundance and Predictive Value of Interferon Stimulated Genes

The relative abundance of ISG15, MX2 and OAS1 mRNA was assessed by RT-qPCR from RNA isolated using the PAXgene blood RNA isolation kit, and our data would tend to support previous observations [10,16,17]. We observed all ISG abundance was elevated due to pregnancy; however, only ISG15 showed a statistical trend of being elevated in pregnant compared to open animals on d 18 of gestation. This trend is somewhat in contrast to an earlier report in which ISG15 did not differ due to pregnancy or pregnancy by day [17]. ROC analysis evaluating the accuracy in ISG relative abundance indicated that OAS1 and MX2 had some predictive value in the identification of pregnant animals; however, ISG15 had none, which is consistent with findings of Green et al., 2010 [17]. Pohler et al., 2017 [8], reported that the detection of ISG15 by end-point PCR and slab gel electrophoresis alone was sufficient to determine pregnancy, as all pregnant animals were positive for the product. In stark contrast, our data, using the same primer sequences for all ISG, each ISG primer combination produced a product present in all samples regardless of pregnancy status or day of gestation (Figure 2). Differences for this observation could be in the procedures to isolate the RNA or the reaction conditions used to generate the product. Pohler et al., 2017 [8], isolated RNA from an enriched white blood cell fraction versus total RNA isolated from blood using the PAXgene system described here. RT-qPCR has typically been interpreted to be more sensitive than end-point PCR; however, this is often not the case. RT-qPCR is the better method for quantification due to real time assessment of product abundance increasing with each cycle [50]. Our data would support this statement, as all three ISG genes were detected using both end-point and RT-qPCR using the same starting mass. Considering our results were similar to, or superior to, detecting ISG gene expression based on both end-point and RT-qPCR, and that other investigators relied heavily on the use of the Trizol protocol, these data indicate that the PAXgene system is sufficient for total RNA isolation for gene detection in bovine blood; however, ISG relative abundance was too inconsistent for verification of pregnancy at d 18. 

### 4.4. Relative Abundance and Predictive Value of Circulating miRNA

Relative abundance of maternal circulating miRNA was evaluated using two separate RNA isolation procedures. The miRNA chosen were previously reported to be present [6,8,29] and, in most instances, differentially expressed in maternal circulation during early gestation. Our BestKeeper analysis is consistent with a previous report that bta-miR-93 and -127 are useful for normalizing miRNA qPCR data generated using bovine serum samples [39]; however, the lack of bta-miR-127 increased abundance due to pregnancy status is in contrast to an earlier report [8] and bta-miR-127 was used in the normalization procedures. The detection of bta-Let-7a in serum is consistent with an earlier report of Ioannidis and Donadeu, 2017 [29]; however, we only examined the relative abundance of one member of the gene cluster they evaluated, and no alternation in the relative abundance of bta-Let-7a due to the day of gestation or pregnancy status was observed. Relative abundance for bta-miR15b, -25, -26 and Let-7a were all elevated in the PAXgene samples compared to their abundance in serum samples. Only bta-miR-16a exhibited similar abundance between PAXgene-isolated RNA and total RNA isolated from serum using Qiagen prep (Table 3). These data indicate (1) the majority (if not all) bta-miR-16a present is in the cell-free maternal circulation, and (2) any cell lysis would result in inaccurate elevated levels for the other miRNA evaluated, as they are more abundant in cells within maternal circulation. Unfortunately, our current experimental design did not determine if bta-miR-16 was associated with extracellular vesicles or with ribonucleotide protein particles. One previous study identified bta-miR-16a abundance increased from d 0 to d 60 of gestation [29] and, in contrast, Pohler et al., 2017, reported bta-miR-16a to be elevated in serum isolated from cows undergoing embryonic loss at d 17 of gestation [8]. Ioannidis and Donadue, 2017 [29], isolated total RNA from plasma, and Pohler et al., 2017 [8], isolated RNA from extracellular vesicles present in plasma. In our study, the only miRNA evaluated that was associated with pregnancy was the tendency for repressed abundance of bta-miR-16a in pregnant samples compared to open samples, regardless of RNA isolation method. Bta-miR-25 tended to be increased in relative abundance due solely to day of gestation, being higher at d 18 within the samples isolated using PAXgene. None of the ROC for the miRNA evaluated were predictive of pregnancy. These data show that source of RNA matters in evaluating miRNA abundance and that none of the evaluated miRNA were predictive of pregnancy status.

## Figures and Tables

**Figure 1 genes-14-01532-f001:**
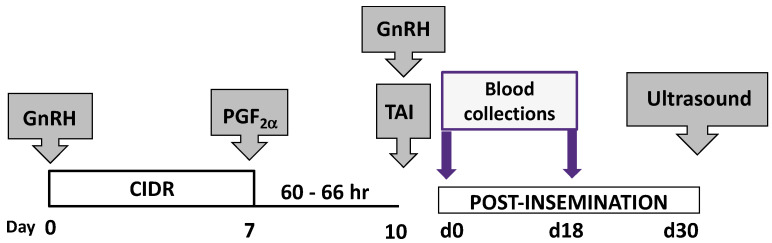
Schematic diagram of the synchronization program, timed artificial insemination, blood sampling and pregnancy detection used. The 7-day Co-synch with CIDR procedure was used for timed artificial insemination (TAI) using semen from a single Angus sire. Day 0 is CIDR insertion and the first injection of GnRH. CIDR were removed at day 7 and 25 mg of PGF2_α_ given i.m. Insemination was conducted 60 to 66 h post-CIDR removal, cows received a second GnRH injection, and blood samples collected at TAI (d 0 post-insemination), d 18, and at d 30. The d 30 sample was used for pregnancy determination by detecting the relative abundance of PSPb. Pregnancy was cross-verified using transrectal ultrasonography.

**Figure 2 genes-14-01532-f002:**
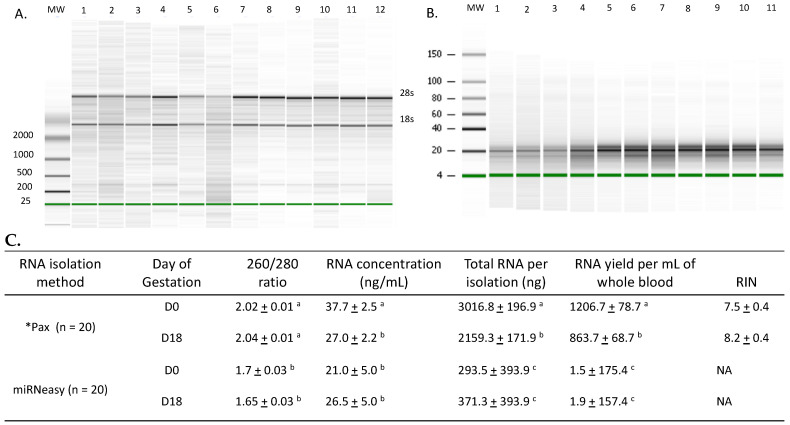
Quality of total RNA collected from whole bovine blood and total RNA isolated from serum. Total RNA isolated from whole blood (n = 20) using the PAXgene blood RNA kit procedure was subjected to quantification, purity and quality analysis using the Agilent 2100, nano 6000 chip (panel (**A**)) spectrophotometry with the Nanodrop 1000 (panel (**C**)). In panel (**A**), relatively equal mass of all samples (determined using the spectrophotometry with the Nanodrop 1000) were run on the Nano 6000 chips. Molecular weight standard sizes are shown to the left. Lane identity is given at the top of the figure and the 28s and 18s ribosomal RNA bands are given to the right. Samples represent cows identified as pregnant (n = 6; lanes 3, 4, 5, 8, 9 and 11) and open (n = 6; lanes 1, 2, 6, 7, 10 and 12). In panel (**B**), total RNA isolated from serum using the miRNeasy procedure (n = 20) was subjected to analysis using the 2100, small RNA chip and spectrophotometry using the Nanodrop 1000 (panel (**C**)). Molecular weight standard sizes are indicated to the left and sample identification is given at the top. Samples represent cows identified as pregnant (n = 6; lanes 3, 4, 5, 8, 9 and 11) and open (n = 5; lanes 1, 2, 6, 7, 10). The percentage of miRNA in the samples did not differ due to treatment or day and all samples exhibited a percentage of miRNA greater than or equal to 10% of the total mass. All parameters indicate RNA of sufficient purity, quantity and quality for further analysis using PCR; however, samples from PAXgene procedures exhibited superior purity, concentration, yield and total mass compared to samples isolated from serum using Qiagen procedures. Values within column containing different ^a, b, c^ superscripts are different (*p* < 0.05).

**Figure 3 genes-14-01532-f003:**
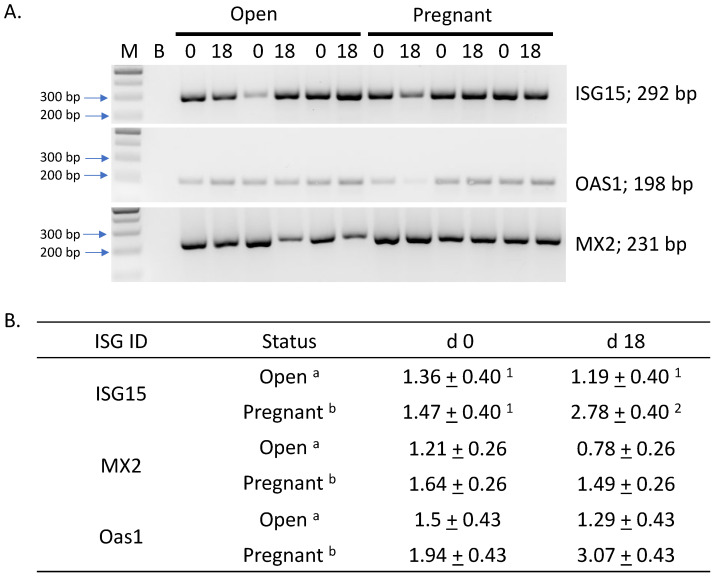
Detection and relative abundance of three INF-stimulated genes using total RNA isolated from maternal whole blood. RT-qPCR was conducted for relative abundance of ISG15, MX2 and OAS1 on d 0 and 18 of pregnancy in cows determined pregnant or open at d 30 of gestation. Panel (**A**) depicts slab gel electrophoresis of each all ISG examined. Product identity and size are given to the right. RT-qPCR (panel (**B**)) depict the relative fold difference in mRNA abundance for ISG15, MX2 and OAS1, respectively. The ^a, b^ superscript present in the status column denotes significance (*p* < 0.05) due to pregnancy, and ^1, 2^ superscript denotes a tendency for pregnancy by day of gestation interaction.

**Figure 4 genes-14-01532-f004:**
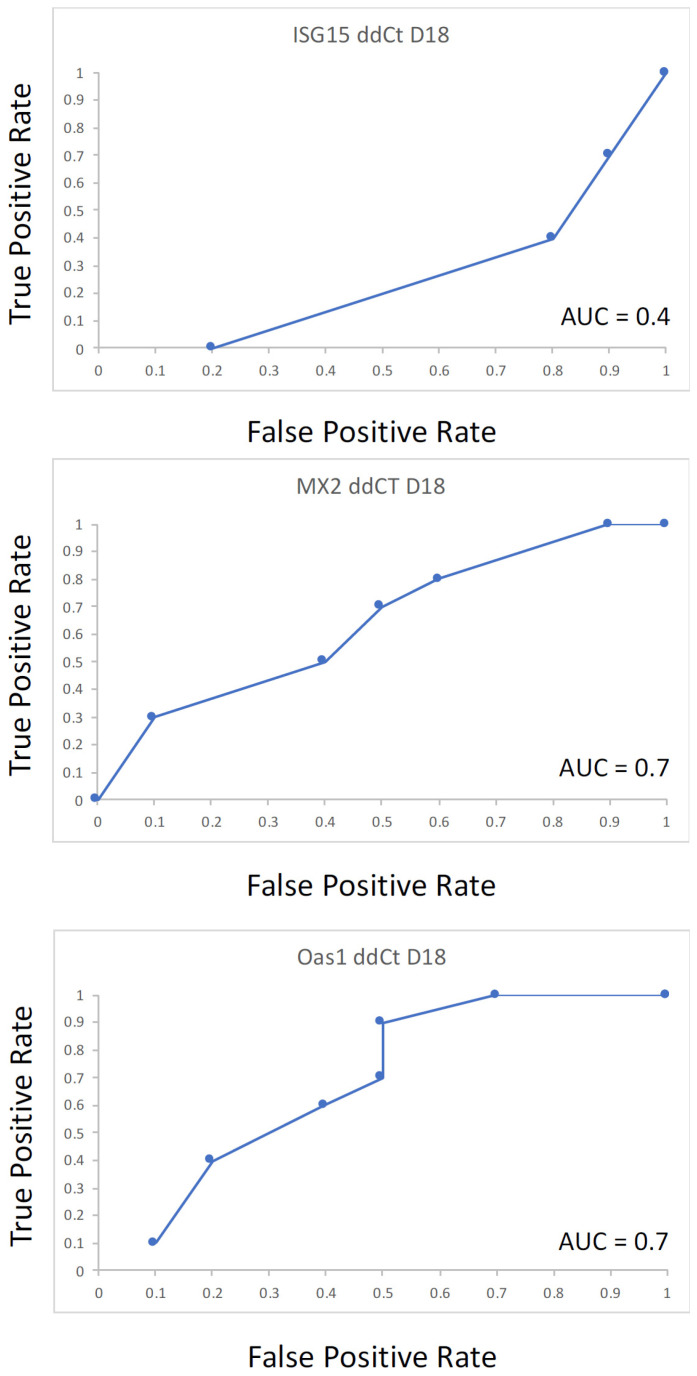
Receiver operator characteristic analysis for three interferon-stimulated genes using the ddCt for the differential abundance on d 18 compared to d 0 of gestation. Receiver operator characteristic curves (ROC) are useful in evaluating performance of diagnostic tests and plot true positive rate (*y*-axis) against false positive rate (*x*-axis). The specific ISG ddCT is shown for each figure at the top and the calculated area under the curve is given in the lower right of each graph.

**Table 1 genes-14-01532-t001:** Accession number, primer sequence and primer efficiency of Interferon-stimulated genes assessed by RT-qPCR.

Gene	Accession Number	Forward Primer Sequence	Reverse Primer Sequence	R2	Slope	Primer Efficiency	Product Size (bp)
Cyclophilin	NM_178320	5′-CACCGTGTTCTTCGACATCG	5′-ACAGCTCAAAAGAGACGCGG	0.93	−3.37	1.98	60
MX2	NM_173941	5′-CTTCAGAGACGCCTCAGTCG	5′-TGAAGCAGCCAGGAATAGTG	0.99	−3.6	1.9	232
Oas1	NM_001040606	5′-ACCCTCTCCAGGAATCCAGT	5′-GATTCTGGTCCCAGGTCTGA	0.99	−2.9	2.21	198
Isg15	NM_174366	5′-CAGCCAACCAGTGTCTGCAGAGA	5′-CCAGGATGGAGATGCAGTTCTGC	0.99	−3.17	2.07	292

**Table 2 genes-14-01532-t002:** The identity, mature sequence and specific Taqman assay for miRNA detected by RT-qPCR.

microRNA	miRNA Identified in Bovine Serum	Chromosome Location	Mature Sequence (5′ to 3′)	miRbase Accession Number	Taqman Assay ID ^#^
miR-25 *	[6,8]	chr25: 37072717-37072800 [+]	CAUUGCACUUGUCUCGGUCUGA	MI0005067	000403
miR-15b *	[8]	chr1: 108321741-108321838 [-]	UAGCAGCACAUCAUGGUUUACA	MI0005012	000390
miR16 *	[8,29]	chr12: 19660386-19660474 [-]	UAGCAGCACGUAAAUAUUGGUG	MI0009753	007149-mat
Let-7a	[29]	chr1: 19984844-19984927 [-]	UGAGGUAGUAGGUUGUAUGGUU	MI0005454	000379
miR-26a *	[6,29]	chr22: 11513974-11514063 [+]	UUCAAGUAAUCCAGGAUAGGCU	MI0009784	000405
miR-127 *	[8,39]	chr21: 67708504-67708598 [+]	UCGGAUCCGUCUGAGCUUGGCU	MI0005008	000452
miR-128	[29]	chr2: 62228129-62228210 [-]	UCACAGUGAACCGGUCUCUUU	MI0004755	002216
miR-93	[39]	chr25: 37072517-37072593 [+]	CAAAGUGCUGUUCGUGCAGGUA	MI0005050	007615_mat

* Denotes that miRNA were reported to have altered abundance due to pregnancy status; ^#^ Taqman probes were checked for specificity to bovine miRNA sequences.

**Table 3 genes-14-01532-t003:** Detection and relative abundance of five miRNA using total RNA isolated from maternal whole blood or maternal serum.

		PaxGene		Qiagen	
miR ID	Status	d 0	d 18	d 0	d 18
Bta-Let-7a *	Open	5.36 ± 1.71	7.43 ± 1.71	0.58 ± 1.71	0.718 ± 1.63
	Pregnant	4.49 ± 1.71	7.73 ± 1.71	1.61 ± 2.04	0.89 ± 2.04
Bta-miR-16a	Open	3.09 ± 0.89	1.98 ± 0.89	2.47 ± 0.81	0.32 ± 0.85
	Pregnant	3.16 ± 0.89	2.47 ± 0.89	4.47 ± 0.99	2.06 ± 0.99
Bta-miR-15b *	Open	13.08 ± 2.82 ^1^	10.38 ± 2.82 ^2^	0.39 ± 2.57 ^1^	0.53 ± 2.57 ^1^
	Pregnant	16.13 ± 2.82 ^1^	10.18 ± 2.82 ^2^	0.80 ± 3.14 ^1^	0.36 ± 3.14 ^1^
Bta-miR-25 *	Open	4.12 ± 6.29 ^a^	22.33 ± 6.29 ^b^	5.77 ± 5.74 ^a^	2.78 ± 5.74 ^a^
	Pregnant	8.95 ± 6.29 ^a^	18.21 ± 6.29 ^b^	4.53 ± 7.03 ^a^	3.25 ± 7.51 ^a^
Bta-miR-26 *	Open	6.73 ± 2.02	6.34 ± 2.02	0.93 ± 1.85	1.32 ± 1.85
	Pregnant	8.03 ± 2.02	9.45 ± 2.02	1.69 ± 2.26	1.18 ± 2.26

Note: RNA was isolated using either maternal whole blood with the PaxGene (PreAnalytiX, Hombrechtikon, Switzerland) procedure or from serum using Qiagen miRNeasy (Qiagen, Ann Arbor, MI) with method indicated below each graph. The relative abundance was determined using the Vandesomplele method was used as shown in the formula: relative gene expression = the RQ _gene of interest_/Geometric mean (RQ _of reference genes_) [40]. ***** indicates differences in relative gene abundance of each miRNA due to RNA source/isolation method (*p* < 0.05). ^1, 2^ superscripts indicate tendencies for significance (*p* < 0.1) RNA source by day of gestation interactions. ^a, b^ super scripts indicate tendencies for significance (*p* < 0.1) for pregnancy status.

## Data Availability

Data available on request.

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
