# Peer review of "Assessing the Differential Abundance of Maternal Circulating MicroRNAs or Interferon-Stimulated Genes with Early Pregnancy"

_genes, 2023, doi:10.3390/genes14081532_

Round 1

Reviewer 1 Report

The paper "Assessing the differential abundance of maternal circulating microRNAs or interferon-stimulated genes with early pregnancy" aimed to assess specific ISG and miRNA abundance on d 18 of gestation". The authors concluded that “These data indicate that ISG show promise as early indicators of pregnancy in cattle, but abundance of the evaluated miRNA do not". (as reported in the abstract).

In those terms paper is not original, as others have measured ISGs expression and associated with pregnancy outcomes  as well as for the miRNA studies, as the Authors rightly reported in the introduction.

However, In the introduction the Authors better specified the objective of the work that was to verify 1) relative ISG mRNA abundance in maternal serum at d 18 of gestation and evaluate their association with pregnancy at d 30 of gestation and 2) determine if specific miRNA previously reported to be present in maternal circulation and associated with pregnancy status differed in relative abundance between d 0 and d 18 of gestation using two different RNA isolation procedures. 

Thus, I understood the novelty of the work is the utilization of Pax blood collection tube and the comparison of two different RNA isolation procedure: from serum or from whole blood collected (using PAX blood collection tubes), as well specified in last part of the manuscript in the discussion paragraph:

“In addition, whether miRNA are isolated from serum, plasma or exosomes, care should be taken to assess the degree of cellular lysis in samples are this could alter any interpretation of relative abundance of specific miRNA. We decided to utilized two procedures for the isolation of RNA for determining relative abundance for the expression of either ISG or miRNA in maternal circulation at d0 and d18 of gestation. Due to the remote location of the cattle available for use in this study, it was decided to utilize the PAXgene blood RNA kit (Pre AnalytiX, Hombrechtikon, Switzerland) as it would allow for stable transport from the animal facility to labs capable of processing the samples for total RNA isolation. These samples would contain total RNA from all blood cells and miRNA in maternal circulation. We also decided to evaluate total circulating miRNA in serum samples and isolated RNA using the Qiagen miRNeasy (Qiagen, Ann Arbor, MI). Efforts to carefully characterize all RNA to ensure quality of the samples are shown in Figure 2.”

All this part has to be moved at the beginning of the manuscript, in part in the introduction (the reason for using a different method for isolation) and in part in the materials and methods.

So the work is more about the methodology and this aim has to be better highlighted, otherwise the study brings no new to what is already reported in the lecterature regarding ISGs or miRNA as early markers of pregnancy.

Author Response

The paper "Assessing the differential abundance of maternal circulating microRNAs or interferon-stimulated genes with early pregnancy" aimed to assess specific ISG and miRNA abundance on d 18 of gestation". The authors concluded that “These data indicate that ISG show promise as early indicators of pregnancy in cattle, but abundance of the evaluated miRNA do not". (as reported in the abstract).

In those terms paper is not original, as others have measured ISGs expression and associated with pregnancy outcomes  as well as for the miRNA studies, as the Authors rightly reported in the introduction.

However, In the introduction the Authors better specified the objective of the work that was to verify 1) relative ISG mRNA abundance in maternal serum at d 18 of gestation and evaluate their association with pregnancy at d 30 of gestation and 2) determine if specific miRNA previously reported to be present in maternal circulation and associated with pregnancy status differed in relative abundance between d 0 and d 18 of gestation using two different RNA isolation procedures. 

Thus, I understood the novelty of the work is the utilization of Pax blood collection tube and the comparison of two different RNA isolation procedure: from serum or from whole blood collected (using PAX blood collection tubes), as well specified in last part of the manuscript in the discussion paragraph:

“In addition, whether miRNA are isolated from serum, plasma or exosomes, care should be taken to assess the degree of cellular lysis in samples are this could alter any interpretation of relative abundance of specific miRNA. We decided to utilized two procedures for the isolation of RNA for determining relative abundance for the expression of either ISG or miRNA in maternal circulation at d0 and d18 of gestation. Due to the remote location of the cattle available for use in this study, it was decided to utilize the PAXgene blood RNA kit (Pre AnalytiX, Hombrechtikon, Switzerland) as it would allow for stable transport from the animal facility to labs capable of processing the samples for total RNA isolation. These samples would contain total RNA from all blood cells and miRNA in maternal circulation. We also decided to evaluate total circulating miRNA in serum samples and isolated RNA using the Qiagen miRNeasy (Qiagen, Ann Arbor, MI). Efforts to carefully characterize all RNA to ensure quality of the samples are shown in Figure 2.”

All this part has to be moved at the beginning of the manuscript, in part in the introduction (the reason for using a different method for isolation) and in part in the materials and methods.

So the work is more about the methodology and this aim has to be better highlighted, otherwise the study brings no new to what is already reported in the lecterature regarding ISGs or miRNA as early markers of pregnancy.

The following changes were made to address the above concerns:

Introduction

Regardless if  miRNA are isolated from serum, plasma or exosomes, care should be taken to assess the degree of cellular lysis in samples are this could alter any interpretation of relative abundance of specific miRNA.  The objective of this study was to verify 1) relative ISG mRNA abundance in maternal serum at d 18 of gestation and evaluate their association with pregnancy at d 30 of gestation and 2) determine if specific miRNA previously reported to be present in maternal circulation and associated with pregnancy status differed in relative abundance between d 0 and d 18 of gestation using two different RNA isolation procedures.  To achieve these objectives, we utilized two procedures for the isolation of RNA for determining relative abundance for the expression of either ISG or miRNA in maternal circulation at d 0 and d 18 of gestation to evaluate the impact of sample isolation procedures on the target relative abundance. Due to the remote location of the cattle available for use in this study, it was decided to utilize the PAXgene blood RNA kit (Pre AnalytiX, Hombrechtikon, Switzerland) as it would allow for stable transport from the animal facility to labs capable of processing the samples for total RNA isolation. These samples would contain total RNA from all blood cells and miRNA in maternal circulation. We also decided to evaluate total circulating miRNA in serum samples and isolated RNA using the Qiagen miRNeasy (Qiagen, Ann Arbor, MI).

Discussion

Therefore we utilized RNA isolated from whole blood or serum to assess miRNA abundance.  It was hoped that the abundance and presence of specific ISG would assist us to identify animals that were pregnant or open at d 18 of gestation based on previous reports.  The PAXgene blood RNA kit (PreAnalytiX, Hombrechtikon, Switzerland) was used to isolate RNA from whole blood as it would allow for stable transport from the animal facility to labs capable of processing the samples for total RNA isolation.  Efforts to carefully characterize all RNA to ensure quality of the samples are shown in Figure 2. 

Reviewer 2 Report

In part 2. Materials and Methods

2.1. Animals and experimental design

Lines 89 and 90 the authors wrote. “Blood samples were obtained on each animal via venipuncture at TAI (day = 0), d 18 post-AI, and d 30 post-AI using two different procedures”.

However, the actual situation was, "blood samples were obtained by venipuncture at three different times, and then the blood was processed using two different methods. Please consider changing the paragraph.

In Figure 1: "Schematic diagram of the synchronization program...." the gray bar representing "7-day Co-synch + CIDR" should be 7 days and not 10 as depicted. In the figure legend (line 107), it says "All cows were inseminated"....... However, all females have been inseminated at a fixed time. Please consider modifying the figure.

The authors note that Blood samples were obtained on each animal via venipuncture at TAI (day = 0), d 18 post-AI, and d 30 post-AI. Also, as far as can be deduced from the manuscript, on day 30, samples and ultrasonography were used to determine the positive or negative gestation of the experimental groups. Thus, day 0 and 18 samples were used to isolate and measure RNA. What criteria did the authors use to affirm that the females negative (non-pregnant) on day 30 were also negative on day 18?

Line 19: The plural verb show does not appear to agree with the singular subject ISG. Please, consider changing the verb form for subject-verb agreement.

Line 20: It seems that the verb do does not agree with the subject. Please, consider changing the verb form.

Line 64: The word aborting doesn’t seem to fit this context. Please, consider replacing it with the correct one.

Line 94: The noun phrase degree seems to be missing a determiner before it. Please, consider adding an article.

Line 100: “to manufacturer’s guidelines”. It seems that there is an article usage problem here.

Line 108: “collections taken at TAI”. It seems that you are missing a verb. Please, consider adding it.

Line 117: “mHz”. Please consider replacing it with the correct one.

Line 134: The noun phrase relative abundance seems to be missing a determiner before it. Please, consider adding an article.

Line 145: The noun phrase equal mass seems to be missing a determiner before it. Please, consider adding an article.

Line 186: The verb were does not seem to agree with the subject. Please, consider changing the verb form.

Line 191: It appears that the preposition from is redundant. Please, consider removing it.

Line 197: “due to day of gestation”. It seems that there is an article usage problem here.

Line 315: It appears that the verb utilized should be in the base form as part of the to-infinitive following decided. Please, consider changing the verb form.

Line 352: The noun phrase identification seems to be missing a determiner before it. Please, consider adding an article.

Line 353: “that detection of”. It seems that there is an article usage problem here.

Line 375: The word consisted doesn’t seem to fit this context. Please, consider replacing it with a different one.

Line 388: The word cleary doesn’t seem to fit this context. Please, consider replacing it with a different one.

Line 395: “from plasma where Pohler…”It seems that conjunction use may be incorrect here.

Author Response

In part 2. Materials and Methods

2.1. Animals and experimental design

Lines 89 and 90 the authors wrote. “Blood samples were obtained on each animal via venipuncture at TAI (day = 0), d 18 post-AI, and d 30 post-AI using two different procedures”.

Phrasing was corrected.

However, the actual situation was, "blood samples were obtained by venipuncture at three different times, and then the blood was processed using two different methods. Please consider changing the paragraph.

Phrasing was corrected.

In Figure 1: "Schematic diagram of the synchronization program...." the gray bar representing "7-day Co-synch + CIDR" should be 7 days and not 10 as depicted. In the figure legend (line 107), it says "All cows were inseminated"....... However, all females have been inseminated at a fixed time. Please consider modifying the figure.

Figure 1 was altered as suggested.  In addition the phrase all cows were inseminated was removed and text edited.

The authors note that Blood samples were obtained on each animal via venipuncture at TAI (day = 0), d 18 post-AI, and d 30 post-AI. Also, as far as can be deduced from the manuscript, on day 30, samples and ultrasonography were used to determine the positive or negative gestation of the experimental groups. Thus, day 0 and 18 samples were used to isolate and measure RNA. What criteria did the authors use to affirm that the females negative (non-pregnant) on day 30 were also negative on day 18?

It was our hopes that the ISG genes would determine pregnancy at d 18 of gestation; however, the accuracy was not reliable as shown in the ISG mRNA abundance data (figure 3) and The ROC curves (Figure 5 now Figure 4).  To clarify, the line “It was hoped that the abundance and presence of specific ISG would allow us to identify animals that were pregnant or open at d 18 of gestation based on previous reports” was added (line 317).

The line “however ISG relative abundance was too inconsistent for verification of pregnancy at d 18.” Was also added (line 371) 

Comments on the Quality of English Language

Line 19: The plural verb show does not appear to agree with the singular subject ISG. Please, consider changing the verb form for subject-verb agreement.

Changed to may serve.

Line 20: It seems that the verb do does not agree with the subject. Please, consider changing the verb form.

Changed to does not.

Line 64: The word aborting doesn’t seem to fit this context. Please, consider replacing it with the correct one.

The phrase correct one is not defined.  However, the word was changed to termination.

Line 94: The noun phrase degree seems to be missing a determiner before it. Please, consider adding an article.

The word the was added

Line 108: “collections taken at TAI”. It seems that you are missing a verb. Please, consider adding it.

Changed to blood samples collected

Line 117: “mHz”. Please consider replacing it with the correct one.

Changed to MHz

Line 134: The noun phrase relative abundance seems to be missing a determiner before it. Please, consider adding an article.

Added the word the

Line 145: The noun phrase equal mass seems to be missing a determiner before it. Please, consider adding an article.

The word an was added

Line 186: The verb were does not seem to agree with the subject. Please, consider changing the verb form.

Changed to was

Line 191: It appears that the preposition from is redundant. Please, consider removing it.

Deleted the word form.

Line 197: “due to day of gestation”. It seems that there is an article usage problem here.

Changed to read Within the RNA isolation method, no difference was observed for quantity or quality of RNA due to day of gestation (Figure 2)

Line 315: It appears that the verb utilized should be in the base form as part of the to-infinitive following decided. Please, consider changing the verb form.

Changed to We utilized two procedures for the isolation of RNA for determining relative abundance for the expression

Line 352: The noun phrase identification seems to be missing a determiner before it. Please, consider adding an article.

Changed to the identification

Line 353: “that detection of”. It seems that there is an article usage problem here.

Changed to the detection

Line 375: The word consisted doesn’t seem to fit this context. Please, consider replacing it with a different one.

Changed to consistent

Line 388: The word cleary doesn’t seem to fit this context. Please, consider replacing it with a different one.

Word cleary deleted.

Line 395: “from plasma where Pohler…”It seems that conjunction use may be incorrect here.

Changed the word to and.

Reviewer 3 Report

The authors aimed 1) to verify relative ISG mRNA abundance in maternal serum at d 18 of gestation and evaluate their association with pregnancy at d 30 of gestation and 2) to determine if specific miRNA previously reported to be present in maternal circulation and associated with pregnancy status differed in relative abundance between d 0 and d 18 of gestation using two different RNA isolation procedures.

Major issues.

The authors do not describe the procedure for selection of animals and for inclusion in the study. This is a serious and significant omission. The authors must describe in detail the procedures and the criteria employed.

Minor issues.

The results can be visualized better. Also, more use of tables should be made to present the findings, which will allow better grasp by future readers.

The discussion is very long and does not help the reader to assimilate the details of the study. I suggest to divide this section in two or three subsections.

Overall.

Re-review after extensive changes as described above.

Author Response

Major issues.

The authors do not describe the procedure for selection of animals and for inclusion in the study. This is a serious and significant omission. The authors must describe in detail the procedures and the criteria employed.

While the information is included in the manuscript, we felt the addition of the following sentence would help for clarification in section 2.2.

A subset of 10 pregnant and 10 open animals at day 30 of gestation were utilized for experiments.  The subset of animals were all of similar age and body condition, isolated serum samples were free of hemolysis, and pregnancy verified by both ultrasonography and the BioPryn ELISA. (line 122)

Minor issues.

The results can be visualized better. Also, more use of tables should be made to present the findings, which will allow better grasp by future readers.

Differential abundance is now reported in tabular form (figure 3 and table 3)

The discussion is very long and does not help the reader to assimilate the details of the study. I suggest to divide this section in two or three subsections.

The discussion was broken into 4 sections.

Round 2

Reviewer 2 Report

The manuscript could be accepted in present form 

Reviewer 3 Report

The authors have improved the manuscript by addressing all the comments made and all the issues raised.